# Eye-Movement Deficits in Seniors with Hearing Aids: Cognitive and Multisensory Implications

**DOI:** 10.3390/brainsci12111425

**Published:** 2022-10-24

**Authors:** Martin Chavant, Zoï Kapoula

**Affiliations:** 1IRIS Laboratory, Neurophysiology of Binocular Control and Vision, CNRS UAR 2022, University of Paris, 45 rue des Saints-Pères, 75006 Paris, France; 2Orasis-Eye Analytics and Rehabilitation, 45 rue des Saints-Pères, 75006 Paris, France

**Keywords:** presbycusis, hearing aids, oculomotricity, cognition, multisensoriality

## Abstract

In recent years, there has been a growing body of literature highlighting the relationship between presbycusis and consequences in areas other than hearing. In particular, presbycusis is linked to depression, dementia, and cognitive decline. Among this literature, the effect of hearing aids, currently the most common method of treating presbycusis, is also a growing research topic. This pilot study aims to explore the effects of hearing aids on the cognitive and multisensory consequences of presbycusis. To that purpose, saccades and vergences eye movements were studied, towards visual and audiovisual targets, of a presbycusis population wearing hearing aids for an average of two years. These measurements were done whether or not participants were wearing their hearing aids. Eye-movement characteristics, particularly latencies (the reaction time taken to initiate an eye movement), allows one to measure attentional and multisensory characteristics. Previous studies showed that presbycusis was linked with an increase of saccade latencies and an improvement in audiovisual interaction capacities, i.e., latencies for audiovisual targets are shorter than those for visual targets. Eye movements are measured and analyzed with REMOBI and AIDEAL technologies. Results show a shortening, with hearing aids, of right saccade latencies to visual targets, suggesting an increase in attention and/or engagement. Yet, saccade latencies are not shorter for audiovisual vs. visual targets alone, neither when wearing hearing aids, nor without. Moreover, convergence latencies are particularly slow for any type of target and with or without hearing aids. The results suggest deficits for audiovisual interactions and the initiation of convergences in that population. These deficits could be part of the factors triggering the need to wear hearing aids. These results therefore show interesting relationships between hearing-aid wearing in a presbycusis population and oculomotricity and invite further research in this area.

## 1. Introduction

Presbycusis is a progressive, age-related hearing loss beginning around the age of 50. Its auditory consequences are a decrease in detection thresholds of high frequencies and a degradation of speech understanding, especially in noise [1]. According to a 2011 study, it affects 63% of people over 70 years of age [2], this prevalence rate is expected to increase in the coming years with the ageing population. The World Health Organization (WHO) estimates that by 2025, more than 500 million people over the age of 60 will have age-related hearing loss [3]. It is a major health problem because in addition to its auditory consequences, fairly recent literature highlights the cognitive and social consequences of presbycusis, namely an increased risk of depression and dementia [4,5] and cognitive decline [6,7,8,9,10] regardless of age. Currently, only the auditory consequences of presbycusis are measured clinically, in order to diagnose and monitor the evolution of presbycusis or to perform auditory rehabilitation. The literature evaluating the link between presbycusis and cognitive decline is still recent, so there are no reference tests to monitor or evaluate its cognitive consequences. In this perspective, it is necessary to develop lines of research that will allow the development of such tests in the future.

Alongside this issue is the question of the effectiveness of hearing aids on these cognitive aspects. Presbycusis is caused in particular by a loss of the inner hair cells and the auditory nerve, allowing the creation of the auditory neural signal [11]. These degradations are irreversible, meaning presbycusis cannot be cured. The most common means of treatment today is the fitting and wearing of hearing aids, numeric devices that amplify and process the sound signal. These devices significantly improve the lives of people with presbycusis [12], although the auditory experience is not perfect and problems with speech comprehension, especially in noise, usually remain. However, their impact on the cognitive consequences of presbycusis has been little studied until now. Studies on this topic are generally not designed for this subject and extract their data retrospectively from large epidemiological cohorts. Of these types of studies, one by Frank and Lin [4], found no association between wearing a hearing aid and a reduced risk of dementia, others have found that wearing a hearing aid is associated with better cognitive scores [13], decreased cognitive decline [7], and improved memory [14]. To our knowledge, two prospective studies have investigated the impact of hearing aids on cognitive performance. A longitudinal study by Castiglione et al. [15] on 125 participants over 65 years found improvement in long-term and short-term memory after hearing rehabilitation. Another longitudinal study (6 months) by Nkyekyer et al. [16] on 40 participants aged 50–90 years found no effect of hearing aids on cognitive scores.

Our previous studies have investigated the links between presbycusis and eye movements allowing us to explore our three-dimensional environment, in order to better understand the cognitive and multisensory consequences of presbycusis. Oculomotricity is an ideal tool to study the mechanisms of action orchestrated by our brain, in particular through these reaction times, associated with attentional cognitive processes [17,18]. Indeed, the latencies of saccades and vergences can be seen as a compact of cognitive executive functions, as they include attentional deployment, fixation release, target localization in space perception, and motor preparation and execution.

The neural networks underlying oculomotor movements have been extensively studied in the past and are among the best-known networks. The study of eye movement has been used in the literature to assess the effects of age, mostly for saccades. Regarding saccades, there is an increase in latencies and a decrease in speed and accuracy with age [19,20,21,22]. This increase of latency with age is also found for the vergence eye movements [23,24,25].

For a population of older adults, these studies have found that age-related hearing loss is associated with (i) an increase in saccade latencies [26] and (ii) an improvement in audiovisual interaction abilities for saccade latencies [27].

The present pilot study was designed to investigate the effect of hearing aids on these relationships. The purpose was to develop this new field of research exploring the cognitive consequences of presbycusis, looking for results concerning the wearing of hearing aids, as well as new methods of measurement to evaluate the cognitive consequences of hearing rehabilitation via hearing aids. To this end, the oculomotor latencies of saccades and vergences, towards visual and audiovisual targets, were studied in a presbycusis population that had been wearing hearing aids for an average of two years. These latencies were compared according to whether or not hearing aids were worn. They were also compared to those of a senior population of the same age who had never worn hearing aids. In order to better target the cognitive processes involved, we also compared these results with performance on the Stroop test, which also measures cognitive executive processes, including selective attention and inhibition [28,29,30,31].

## 2. Materials and Methods

The study was conducted in accordance with the Declaration of Helsinki, and approved by Ethics Committee “Ile de France II” (N° ID RCB: 019-A02602-55, approved the 10 March 2020).

### 2.1. Participants

A group of hearing-aided seniors with presbycusis (group Senior_HA) and a group of seniors who have never worn hearing aids (group Senior) were tested. The group Senior_HA was composed of 21 participants (mean age 75.4 ± 6.7). Participants in this group had been wearing their hearing aids for an average of 2.21 ± 1.5 years. The group Senior was composed of 20 participants (mean age 72.05 ± 8.4). They all completed a short questionnaire at the beginning of the tests to report if they had good vision in their daily life (with or without correction), if they took medications that could damage their sensory and motor functions, or if they had neurologic, psychiatric, auditory, or oculomotor disorders. Those who had cataracts were treated. All the participants had good sight or wore visual correction. During the tests, none of the participants complained about not seeing the visual target. Thus, the participants were considered to be representative of a normal aging population. Informed consent was obtained from all of the participants after the nature of the procedure had been explained. The search for participants was conducted during a period of confinement of the COVID-19 pandemic. Participants were recruited mainly through the cognitive research network RISC and by advertising in associations that may have senior participants.

### 2.2. Hearing Tests

A professional audiometrist was in charge of performing two kinds of audiometry, in a sound booth calibrated cabin, with an audiometer of the brand Interacoustics (model AD639, Paris, France). The outer ear canals of all the participants were all checked before the hearing tests.

#### 2.2.1. The Tonal Audiometry

This test aims to assess the audibility, i.e., the minimum intensity required for a sound to be heard. It was realized, for each ear separately, with the headset TDH-39P. For each ear was determined the lowest intensity in dB HL needed by the participant to detect the following pure tone: 250, 500, 750, 1000, 2000, 3000, and 4000 Hz. The final score is called the Pure Tone Average (PTA) and represents, for each ear, the mean of all these thresholds. Only the best PTA between the two ears was retained to match to the hearing-loss definition of the World Health Organization. They consider a hearing loss when the best PTA between the two ears is bigger than 20 dB HL [32].

#### 2.2.2. The Vocal Audiometry in Silence

This test aims to assess speech comprehension in silence. Lists of words with differing intensities are sent to the participant, enabling them to calculate a comprehension score for a given intensity. It was performed with a loudspeaker situated at 1 m in front of the participant (azimuth 90°). Thus, the two ears were tested simultaneously. The different intensities with which the lists were sent were 70, 60, 50, 40, 30, 20, or 10 dB SPL. The lists used were the Lafon cochlear lists, composed of 17 monosyllabic words of 3 phonemes (51 phonemes) [33]. The comprehension score for each list was calculated with the percentage of recognized phonemes in the list. The final score is called the SRT50 (Speech Recognition Threshold 50%) and represents the intensity required to understand 50% of the phonemes of a list. The SRT50 was estimated by a cross product between the intensities needed for the lists with the score just above and below 50% of comprehension.

The hearing tests were performed with and without hearing aids.

### 2.3. Oculomotor Tests

Divergence, convergence, left saccade, and right saccade were elicited with the REMOBI device, which is a visio-acoustic device developed by our laboratory (Figure 1).

REMOBI (Patent US885 1669, WO2011073288) is a plane surface upon which are disposed red LEDs (frequency 626 nm, 180 mCd, and a diameter of 3 mm). Each LED is equipped with a buzzer delivering a 2048 Hz pure-tone of 70 dB SPL. Participants were sitting and the REMOBI was placed at their eye level. The instructions given to them were to look at the only LED that was on, as quickly and accurately as possible, and then to maintain the fixation as the LED was still on, all of which without moving the head. Thus, the localization and patterns of LEDs allowed the testing of the desired eye movements.

#### 2.3.1. Saccades Sequence

During this sequence were elicited 20 trials of a saccade to the right and 20 trials of saccade to the left, randomly interleaved. For each trial, participants first fixated a central LED, situated at 70 cm in front of them (same distance from his two eyes). The right and left saccades were elicited by lightning a peripheral LED, also at 70 cm, but at 20° to the right or to the left from the central LED.

#### 2.3.2. Vergence Sequence

During this sequence were elicited 20 trials of convergence and 20 trials of divergence, randomly interleaved. All the LEDs were situated in front of the participant (same distance between the left and right eye) and varied only in depth. For each trial, participants first fixated a central LED, situated at 40 cm from them. The divergence and convergence were elicited by lighting a peripheral LED, situated at either 20 cm (Convergence) or 150 cm (Divergence) from the participant.

The normal proximal point of convergence is supposed to be less than 10 cm [34]. It should therefore have been possible for our participants to see the LED target presented at 20 cm for the convergence test; our eye-movement recording confirmed this. The amplitude of the movement achieved during the first 250 ms (including the open loop period of approximately 80 ms and the subsequent 160 ms period, see Section 2.5 Eye-Movement Analysis) was 46 ± 18% for divergence (mixing the results of the two groups, group Senior_HA and group Senior); for convergence it was 51 ± 25%.

For saccade and vergence sequences, the central LED was switched on during a random time between 1200 and 1800 ms. The peripheral LED was lit for 2000 ms. There was an overlap time of 200 ms where the two LEDs were lit at the same time (overlap paradigm). The trials were separated by a blank period of 300 to 700 ms. The total duration of a sequence was approximately 150 s.

The hearing tests were performed with and without hearing aids. The score extracted is the latency, expressed in ms, and representing the time between the activation of the peripheral LED and the initiation of the movement. 

### 2.4. The Targets Modality

The saccade and vergence sequences were tested for visual targets and for audiovisual targets. For the visual targets, the LEDs turned on without activation of their adjacent buzzer. For the audiovisual targets, an auditory signal was sent with the adjacent buzzer of the LED 50 ms before the activation of the LED. This auditory signal was activated for a duration of 100 ms.

### 2.5. Eye-Movement Analysis

The eye movements were captured with the head-mounted video-oculography device, Pupil Core (Pupil Labs, Berlin, Germany), and the data acquired were analyzed with the AIDEAL software (pending international patent application: PCT/EP2021/062224 7 May 2021). It derived the signal by calculating the difference between the two eyes from the individual calibrated eye position signals (i.e., left eye–right eye). The total saccade movement was measured by defining its onset and offset when the velocity of the movement was above or below 10% of its peak velocity. The movement of vergence was divided into two components, following the double mode control of the vergence model. This model divides the dynamic of vergence into two chronological steps: i) an initial step of enhanced speed without visual feedback (open-loop) and then ii) a sustaining step, slower and driven by visual feedback (closed-loop) [35,36,37]. The initial open-loop component was defined by AIDEAL when the velocity of vergence was above 5°/s. The following closed-loop component was measured by extending the analysis of the next 160 ms. Then different filters on trials ware applied. AIDEAL removed first the trials with blinks and outliers, and then, the value greater than the mean plus twice the standard deviation. 

### 2.6. Stroop Test

The French version of Stroop Victoria was used in the current study [30]. It is composed of three parts. In each part, participants enumerate as quickly and accurately as possible the color of 24 items (6 lines of 4 items) of a sheet of A4 format. In the first part, the items are dots, and it is called the Dot Condition. In the second part, the items are irrelevant worlds (world with neutral meaning), and it is called the Word Condition. The items of the third part are color words, irrelevant from their ink impression (the word “blue” written in red). This last part is called the Interference Condition.

The selective attention and inhibition capacities are evaluated by comparing the performances during the Dot and Interference Conditions. The Dot Condition assesses the baseline capacity of color recognition and enumeration. The Interference Condition also assesses these capacities but with the interference of the words. The information given by the instinctive reading of the word is more protruding than its actual color, and it has to be inhibited by the brain.

The scores extracted from the Stroop test were: Stroop_D, representing the time taken to finish the Dot Condition; Stroop_I, representing the time taken to finish the Dot Condition; Stroop_I/D, representing the ratio of the time taken to finish the Dot Condition on the time taken to finish the Interference Condition. Higher Stroop_I/D reveals a lower capacity of inhibition.

## 3. Results

### 3.1. Auditory and Cognitive Characterizations

The results for the hearing scores are grouped in Figure 2. The PTA (measured by tonal audiometry) of all participants of group Senior were above 20 dB HL. According to the WHO criterion, all participants in this group had at least a mild hearing loss. Despite this, their hearing was still better than that of the senior hearing-aided population (group Senior_HA), as the PTA and SRT50 of the group Senior were significantly smaller than those of the participants of the group Senior_HA, with or without their hearing aids.

Wearing hearing aids significantly improved hearing scores, i.e., PTA and SI50. With their hearing aids, the hearing-impaired seniors had a similar PTA as the unimpaired seniors. However, their SI50 was still significantly higher (34.5 ± 8.3 vs. 26.8 ± 10.9, *p* = 0.002).

Concerning the cognitive outcomes, Figure 3 shows the results of the Stroop test. There is no significant difference between the participants of the group Senior_HA and the participants of the group Senior. It should be noted that the participants of the group Senior_HA passed the Stroop test while wearing their hearing aids.

These two populations (with and without hearing aids), similar in age, had similar cognitive performances, but differed in their hearing, especially on the parameter of speech understanding in silence (SRT50).

### 3.2. Oculomotor Latencies and Audiovisual Interactions

#### 3.2.1. Influence of Hearing Aids for the Participants of the Group Senior_HA

First, the latencies of the group Senior_HA are compared when their participants were wearing or not their hearing aids. The results in Figure 4 show only one significant effect: the latency of right saccades to visual targets was shorter when participants wore their hearing aids (*p* = 0.036). However, there was no significant effect of hearing aids for eye-movement latencies to audiovisual targets.

The only impact of wearing hearing aids, in a population with presbycusis and wearing hearing aids for several years, was therefore the shortening of the latency of saccades to the right.

#### 3.2.2. Comparison between the Group Senior_HA and the Group Senior

The results are also shown in Figure 4.

For visual targets, the latency of the convergences was significantly longer for the group Senior_HA, with or without their hearing aids, than for the group Senior (395 ± 70 ms for Senior_HA0 and 403 ± 73 ms for Senior_HA1 vs. 342 ± 69 ms for Senior).

For audiovisual targets, the latency of the right saccades was significantly longer for the group Senior_HA with or without their hearing aids, than for the group Senior (327 ± 73 ms for SA0 and 329 ± 64 ms for SA1 vs. 274 ± 41 ms for SnA_2). The same phenomenon was observed for the leftward saccades, but only when the participants of the group Senior_HA wore their hearing aids (324 ± 58 ms for SA1 vs. 268 ± 51 ms for SnA_2).

There was an audiovisual facilitation for saccades in the group Senior, i.e., a significant reduction in latency when presenting audiovisual targets compared to visual targets (267 ± 51 ms vs. 301 ± 72 ms for leftward saccades; 274 ± 41 ms vs. 318 ± 74 ms for rightward saccades). This result can be considered as the origin of the previous result, showing shorter saccade latencies towards audiovisual targets for the group Senior. This facilitation was not found in the group Senior_HA, with or without their hearing aids.

## 4. Discussion

### 4.1. The Immediate Effect of Hearing Aids on Oculomotricity and Audiovisual Facilitation in a Presbycusis Hearing-Aided Population

#### 4.1.1. Wearing Hearing Aids Increases Concentration

The results suggest a decrease in the latency of right saccades to visual targets with the wearing of hearing aids in a presbycusis hearing-aided population. The auditory amplification of the background noise of the room (not soundproofed for the oculomotor tests) and of the noises coming from the participants (breathing, chewing, swallowing, etc.) could induce a more important state of alertness and concentration, resulting in the reduction of the latency.

There are studies evaluating the influence of background noise on detection tasks or various cognitive tasks. This literature presents numerous contradictory results, depending in particular on the nature of the sound used and the test situation. Some studies have shown an improvement in performance in the presence of background noise. Davies et al. [38] and Poulton [39] suggested that the presence of this noise could improve performance by a mechanism of arousal and concentration; however, this extra concentration could also disappear with habituation. T. Auburn et al. [40] found that loud noise increases the speed of detection of a sequence of two identical digits during the first twenty minutes of a task. Moradi et al. [41] found that selective attention, as measured by the DUAF test, is enhanced by stressful background noise. Helton et al. [42] found that during a 12 min vigilance task, intense noise increases task engagement and detection scores. The duration of the oculomotor tests in our studies was approximately 15 min, corresponding to the period during which the alertness condition affects performance in previous studies [40,41,42]. This result, the shortening of saccade latencies to visual targets with hearing amplification, was also found for a young participant with normal hearing in a case study of Kapoula et al. [43].

It is interesting to note that this result was not found for movements towards audiovisual targets: the wearing of hearing aids no longer accelerated the latency. The sound of the audiovisual target could itself act as an arousal factor. Adding sound may have counterbalanced the effects found with visual targets.

It is interesting to find these results only for rightward saccades, not for leftward ones. This bias may be related to the overtraining of rightward saccades which are the motor basis of reading activity. Wearing the hearing aids would act as a general warning effect improving the latency of the most frequently made saccades to the right.

#### 4.1.2. The Influence of Hearing Aids on Audiovisual facilitation

In the senior hearing-aided population, the difference between the latency of saccades to visual targets and the latency of saccades to audiovisual targets remains the same with or without hearing aids. This result suggests that wearing hearing aids does not improve the audiovisual facilitation of saccade latencies in a presbycusis-impaired population. Given that hearing aids amplify the sound of the audiovisual target, this issue is similar to that of the influence of sound intensity on audiovisual facilitation: does decreasing or increasing the intensity of the sound of an audiovisual target alter the audiovisual facilitation observed on saccade latency?

Few studies in the literature can provide answers concerning this particular context. There are indeed studies assessing multisensory facilitation as a function of the intensity of different sensory modalities [44,45,46,47]. However, it is difficult to compare our results to theirs, particularly because these studies consider responses for all sensory modalities and vary the intensity of all these modalities. In our study, responses to an auditory-only target were not measured. However, a study by Arndt et al. [48] addressed this issue and had a protocol very similar to ours, allowing for a comparison. This study shows an increase in audiovisual facilitation (reflected by a reduction in saccade latency in the presence of audiovisual versus visual targets) with increasing target sound intensity. The sound was a noise with a wide bandwidth (500 to 14,000 Hz) that could vary in intensity over a range of 12 dB; the saccades studied were 25 degrees to the right or left. If, as suggested in the study of Arndt et al. [48], greater sound intensity results in greater audiovisual facilitation, then wearing the devices in the presbycusis-fitted population could have improved audiovisual facilitation of saccades.

It is possible that the amplification via hearing aids of the audiovisual target sound was not significant enough to improve the audiovisual facilitation. This amplification is not exactly known. In the senior hearing-aided population, the average gain given by the hearing aids for the 2000 Hz detection threshold (the frequency tested in the tonal audiometry that most closely resembles the sound of the audiovisual targets) was 16 ± 7 dB. However, hearing aids have different amplification depending on the intensity received. In general, the louder the sound is received by the hearing aids, the less it will be amplified. Since the sound of the audiovisual target is well above the detection thresholds, its amplification via the hearing aids is therefore less than 16 dB on average.

Note also that, in our study, sound appears 50 ms before diode activation. This interval was chosen because it causes the greatest audiovisual facilitation [49,50,51]. However, it could also mask a potential improvement in audiovisual facilitation with increasing sound intensity. Indeed, the study of Arndt et al. [48] estimates that increasing the intensity of the audiovisual target sound reduces saccade latencies by increasing the processing speed of the auditory modality. In other words, the auditory modality of the audiovisual target is integrated earlier when the intensity is higher. A study by Corneil et al. [44] and another by Hughes et al. [47] show, in fact, a decrease in the latency of saccades to an auditory target with increasing target intensity. In our case, the behavioral improvement, given by the earlier integration of the auditory modality as a consequence of the increase in its intensity, could reach a ceiling effect with the 50 ms lead that the auditory modality has.

In summary, our results do not show an improvement of audiovisual facilitation with the wearing of hearing aids, in a presbycusis population wearing hearing aids for several years. However, this issue should be addressed with new paradigms, for example by controlling the amplification of the hearing aids on the audiovisual target, or by simultaneously activating the sound and the diode of the audiovisual targets.

Furthermore, it is very important to note that the senior hearing-aided population seemed to have deficits in audiovisual facilitation. This point is developed below in Section 4.2.

### 4.2. Audiovisual Facilitation and Vergence Slowness as Biomarkers of the Need for Hearing Aids

The results show a significant decrease in saccade latencies for audiovisual versus visual targets in the senior population. This audiovisual facilitation is a well-known result in the literature [44], [52,53,54,55,56]. However, this audiovisual facilitation is absent for the senior hearing-aided population. The main difference between these two populations concerns hearing: the hearing ability of the group Senior_HA is more impaired than that of the group Senior, even with hearing aids, particularly for understanding in silence. Therefore, the deficit in audiovisual facilitation could come from the poorer hearing of the group Senior_HA. However, this interpretation is at odds with the results of a previous study [27], showing an improvement in audiovisual facilitation, i.e., reduction of saccade latency with audiovisual targets versus visual targets, with physiological audibility loss in an elderly population.

Thus, the lack of audiovisual facilitation could be one of the characteristics and reasons why seniors with presbycusis decide to wear hearing aids. The results of our previous study [27] also indicate a large variability in audiovisual facilitation as a function of hearing. Thus, although these results show, on average, an improvement in audiovisual facilitation with hearing loss, they also indicate a substantial number of seniors with both poor hearing and poor audiovisual facilitation. Since audiovisual interaction capacities seem to act as a compensatory system during hearing decline [57], the loss of this multisensory compensation in a person with presbycusis would act as a trigger for hearing aid fitting. This mechanism may have created a selection bias in the group Senior_HA, explaining the low audiovisual interaction capacities.

These results therefore question the role of decreased multisensory abilities as a trigger for the desire for a hearing aid. Audiovisual facilitation for saccade latency could become an indicator of the need for a hearing aid for a person with hearing loss.

Another possible biomarquer is the abonrmal sloweness of vergence initiation (long convergence latency) in such a population present in both testing conditions with or without hearing aids. Reacting rapidly to a target coming towards and closer to a participant is vital and seems to be delayed by about 50 ms relative to the population with no hearing aids. This results is surprising and needs further investigation; a possible interpretation would be a side effect of hearing aids emphasizing the left–right auditoty perception at the expense of the near–far axis.

## 5. Conclusions

This study shows several relationships between auditory rehabilitation in a presbycusis population and their oculomotor and multisensory abilities, leading to various interpretations, and opening the way for further studies on this unexplored topic.

The results suggest an immediate effect of hearing aids on concentration and alertness. There also appear to be deficits in multisensory and convergence initiation abilities in presbycusis patients with hearing aids. These deficits could be part of the factors leading to the willingness to start hearing rehabilitation. It would be interesting to discover the mechanisms modulating these deficits. It is possible that the hearing aid users tested were fitted too late to recover their multisensory compensation abilities.

It is important to remember that these results were extracted from a small cohort. They should be confirmed with more data. In addition, only significant results were reported. We might have missed effects that were only trends in our study that would prove significant with more data. It would also be important to better control for certain variables, such as the amount of time hearing aids were worn on a daily basis, the length of time the participants had been wearing their hearing aids, or the length of time they had been hearing. Indeed, the amount of time worn and the length of time the hearing aids have been in use affect their effectiveness [58,59,60].

## 6. Patents

**REMOBI**: US885 1669, WO2011073288. **AIDEAL**: PCT/EP2021/062224 7 May 2021.

## Figures and Tables

**Figure 1 brainsci-12-01425-f001:**
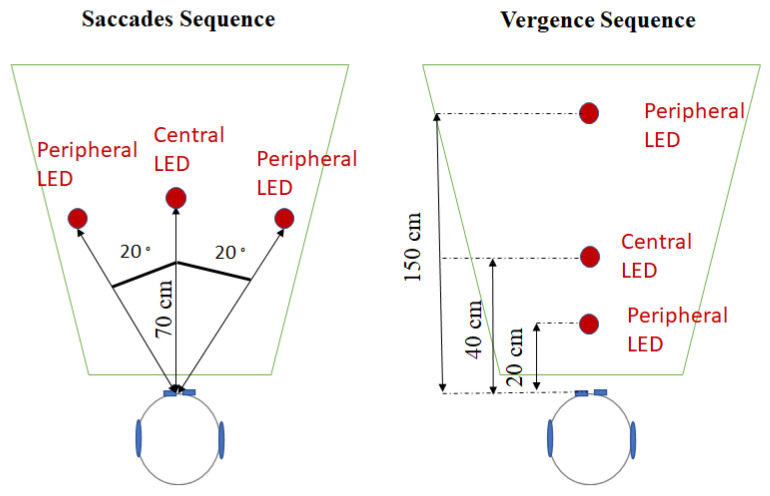
Top-view of the position of the LEDs for the saccades sequence (on the left) and for the vergence sequence (on the right).

**Figure 2 brainsci-12-01425-f002:**
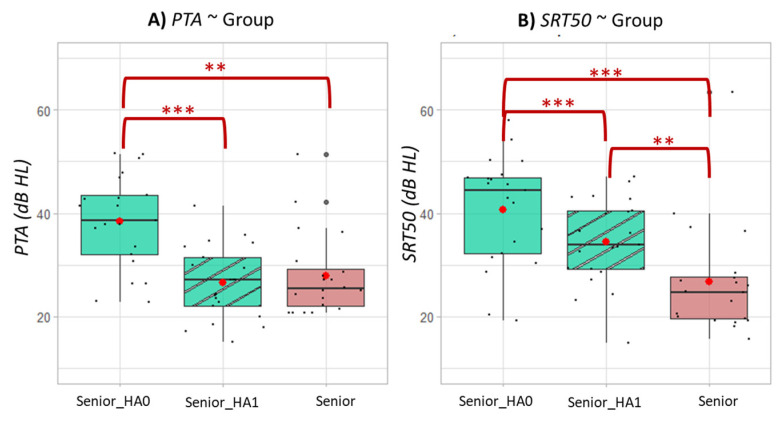
Hearing scores (**A**) for PTA and (**B**) for SRT50 as a function of group. Senior_HA0 represents the group Senior_HA when participants did not wear their hearing aids. Senior_HA1 represents the group Senior_HA when participants were wearing their hearing aids. Senior represents participants the group Senior. Asterisks indicate Wilcoxon test *p* values less than 0.05: ‘**’ for 0.001 < *p* < 0.01; ‘***’ for *p* < 0.001.

**Figure 3 brainsci-12-01425-f003:**
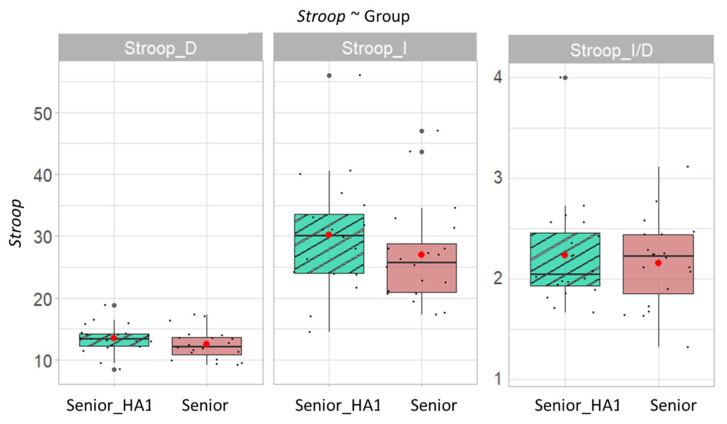
Stroop test scores as a function of group. Senior_HA1 represents the group Senior_HA when participants were wearing their hearing aids. Senior represents participants in the group Senior.

**Figure 4 brainsci-12-01425-f004:**
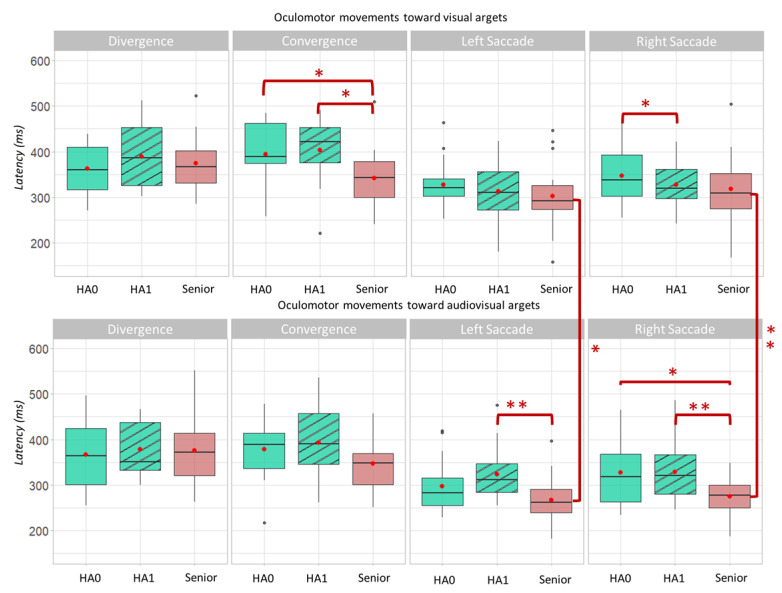
Oculomotor latencies as a function of eye movement, groups (HA0 represents group Senior_HA when participants are not wearing their hearing aids, HA1 represents group Senior_HA when participants were wearing their hearing aids, Senior represents participants in group Senior), and target sensory modality. Figures in the upper part represent eye movements to visual targets, figures in the lower part represent eye movements to audiovisual targets. Asterisks indicate Wilcoxon test *p* values less than 0.05: ‘*’ for 0.01 < *p* < 0.05; ‘**’ for 0.001 < *p* < 0.01.

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
