# Peer review of "Eye-Movement Deficits in Seniors with Hearing Aids: Cognitive and Multisensory Implications"

_brainsci, 2022, doi:10.3390/brainsci12111425_

Round 1

Reviewer 1 Report

Summary

This manuscript reports results from an experimental study with elderly patients, who were wearing hearing aids. The study focused on the effect of the use of hearing aids on the cognitive and multisensory effects of presbycusis through the use of eye movement measures.

Thank you for the opportunity to review this manuscript. I enjoyed reading this manuscript and I think it has considerable potential to contribute to the literature in several ways. It can add further evidence to the literature on better understanding the effects of presbycusis on cognition and the importance of use of eye movement tools for hearing aids research.

There were some parts where text was on the side due to formatting of tables and figures, so that will need editing.

Introduction

Very clear and detailed account of a wide range of studies on the research conducted around the effects of presbycusis on hearing and cognitive skills. I think the very review of the studies on hearing aids and their benefits and effects on a wide range of skills really helped build the rationale for this study. I think it would be beneficial to provide a more comprehensive review of eye movements in elderly population in order to provide a more in-depth foundation as the basis for this study. Further details were needed to help explain the rationale behind the choice to use oculomotricity as the main tool for this study.

Method

In the participants section, please clarify for the reader sampling method and how participants were recruited.

It is also important in my opinion to include a clear and detailed account of the procedure that was followed.

A detailed account of the eye movement measures, and tools has been provided, so that enhanced the clarity of the methods employed.

Results

Very clear and detailed presentation of the statistical analyses and I think the inclusion of the results in figures helps highlight the key findings for the experimental groups and conditions.

Discussion

The authors have provided a comprehensive description of the many findings and an explanation of their findings in the light of previous literature. The results provide an interesting contribution to the existing literature in terms of the effect of presbycusis on cognitive and multisensory skills. The limitations of the study have clearly been considered and the need for further research on this topic has been emphasised. I think a few more details about the key messages from this study in the conclusion would be highly beneficial.

Author Response

Thank you for the report. 
We answer in the blue sections, as well as in the article. 

Reviewer 2 Report

The study examined the latency of saccades and vergence eye movements, as well as inhibitory control in older adults with and without hearing loss. The introduction provides a nice synthesis of the literature on this topic, however, it could be improved by providing a clearer rationale for the hypotheses, why did the authors explore the latency of the saccades and vergence and the Stroop task is not clear. Are these tests assessing different aspects of information processing, are they related?

The methods were clearly described, but the figures obstructed some of the text so I could not read it. Please clarify that the participants were able to converge on the target at 20 cm, this target is very close and I’d expect that older individuals would have some difficulty with accommodating on this target. The description of the expected vergence response is very clear (open vs closed loop), but there was no further mention of these results. If vergence movements were analysed in such detail, please report these findings rather than just the latency.

The authors mentioned that the groups had good vision, it is not clear how that is defined. If available, provide additional details, such as acuity, etc. Most older individuals do not have normal vision, aging is associated with reduced contrast sensitivity, acuity, or stereo and often associated with cataracts, glaucoma or AMD. Were any of these screened out?

It seems that the saccade latency was different with hearing aids compared to no aids but only for the right sided targets. This is unexpected. The rightward bias seems to be in conflict with the attention explanation that the authors suggested because attentional networks are lateralized to the right hemisphere, which would create a leftward bias, ie saccades to left should be faster. Perhaps I misunderstood the authors’ argument, please clarify.

Discussion section 4.1.1 titled ‘Wearing hearing aids increases concentration’ is very interesting but I don’t see which results from this study support such a conclusion. Similarly, for section 4.2, I’m not convinced that this study provides sufficient evidence for audiovisual facilitation to serve as a biomarker of the need for hearing aids.

Author Response

(The authors gave the same response as above.)
